# Evaluating Agents Without Rewards

## Abstract

Reinforcement learning has enabled agents to solve challenging control tasks from raw image inputs. However, manually crafting reward functions can be time consuming, expensive, and prone to human error. Competing objectives have been proposed for agents to learn without external supervision, such as artificial input entropy, information gain, and empowerment. Estimating these objectives can be challenging and it remains unclear how well they reflect task rewards or human behavior. We study these objectives across seven agents and three Atari games. Retrospectively computing the objectives from the agent's lifetime of experience simplifies accurate estimation. We find that all three objectives correlate more strongly with a human behavior similarity metric than with task reward. Moreover, input entropy and information gain both correlate more strongly with human similarity than task reward does.

## 1 Introduction

Deep reinforcement learning (RL) has enabled agents to solve complex tasks directly from high-dimensional image inputs, such as locomotion (Heess et al., 2017), robotic manipulation (Akkaya et al., 2019), and game playing (Mnih et al., 2015; Silver et al., 2017). However, many of these successes are built upon rich supervision in the form of manually defined reward functions. Unfortunately, designing informative reward functions is often expensive, time-consuming, and prone to human error (Krakovna et al., 2020). Furthermore, these difficulties increase with the complexity of the task of interest.

In contrast to many RL agents, natural agents generally learn without externally provided tasks, through intrinsic objectives. For example, children explore the world by crawling around and playing with objects they find. Inspired by this, the field of intrinsic motivation (Schmidhuber, 1991; Oudeyer et al., 2007) seeks mathematical objectives for RL agents that do not depend on a specific task and can be applicable to any unknown environment. We study three common types of intrinsic motivation:

- Input entropy encourages encountering rare sensory inputs, measured by a learned density model (Schmidhuber, 1990; Bellemare et al., 2016b; Pathak et al., 2017; Burda et al., 2018b).

- Information gain, or infogain for short, rewards the agent for discovering the rules of its environment (Lindley et al., 1956; Houthooft et al., 2016; Shyam et al., 2018; Sekar et al., 2020).

- Empowerment measures the agent's influence it has over its sensory inputs or environment (Klyubin et al., 2005; Mohamed and Rezende, 2015; Karl et al., 2017).

| Metric | Reward Correlation |
|---|---|
| Task Reward | 1.00 |
| Human Similarity | 0.67 |
| Input Entropy | 0.54 |
| Information Gain | 0.49 |
| Empowerment | 0.41 |

| Metric | Human Correlation |
|---|---|
| Human Similarity | 1.00 |
| Input Entropy | 0.89 |
| Information Gain | 0.79 |
| Task Reward | 0.67 |
| Empowerment | 0.66 |

Table 1: Correlation coefficients between each metric and task reward or human similarity. The 3 task-agnostic metrics correlate more strongly with human similarity than with task reward. This suggests that typical RL tasks may not be a sufficient proxy for intelligent behavior seen in humans playing the same games.

Despite the empirical success of intrinsic motivation for facilitating exploration (Bellemare et al., 2016b; Burda et al., 2018b), it remains unclear which family of intrinsic objectives is best for a given scenario, for example when task rewards are sparse or unavailable, or when the goal is to behave similarly to human actors. Moreover, it is not clear whether different intrinsic objectives offer

similar benefits in practice or are orthogonal and should be combined. To spur progress toward better understanding of intrinsic objectives, we empirically compare the three objective families in terms of their correlation with human behavior and with the task rewards of three Atari games and Minecraft Treechop.

The goal of this paper is to gain understanding rather than to propose a new intrinsic objective or exploration agent. Therefore, there is no need to estimate intrinsic objectives while the agents are learning, which often requires complicated approximations. Instead, we train several well-known RL agents on three Atari games and Minecraft and store their lifetime datasets of experience, resulting in a total of 2.1 billion time steps and about 9 terabytes of agent experience. From the dataset of each agent, we compute the human similarity, input entropy, empowerment, and infogain using simple estimators with clearly stated assumptions. We then analyze the correlations between these metrics to understand how they relate to another and how well they reflect task reward and human similarity.

The key findings of this paper are summarized as follows:

- Input entropy and information gain both correlate better with human similarity than task reward does. This implies that to measure how similar an agent's behavior is to human behavior, input entropy is a better approximation than task reward.

- Simple implementations of input entropy, information gain, and empowerment correlate well with human similarity. This suggests that they can be used as task-agnostic evaluation metrics when human data and task rewards are unavailable.

- As a consequence of the these two findings, task-agnostic metrics can be used to measure a different component of agent behavior than is measured by the task rewards of the reinforcement learning environments considered in our study.

- Input entropy and information gain correlate strongly with each other, but to a lesser degree with empowerment. This suggests that optimizing input entropy together with either of the two other metrics could be beneficial for designing exploration methods.

This paper is structured as follows. Section 2 describes the games and agents used for the study. Section 3 details the experimental setup and estimators used to implement the metrics. Section 4 discusses quantitative and qualitative results. Section 5 summarizes key take-aways and recommendations.

## 2 BACKGROUND

To validate the effectiveness of our metrics for task-agnostic evaluation across a wide spectrum of agent behavior, we retrospectively computed our metrics on the lifetime experience of well-known RL agents. Thus, we first collected datasets of a variety of agent behavior on which to compute and evaluate our metrics.

**Environments** We evaluated our agents in three different Atari environments provided by Arcade Learning Environment (Bellemare et al., 2013): Breakout, Seaquest, and Montezuma's Revenge, and additionally in the Minecraft Treechop environment provided in MineRL (Guss et al., 2019). Breakout and Seaquest are relatively simple reactive environments, while Montezuma's Revenge is a challenging platformer requiring long-term planning. Treechop is a 3D environment in which the agent receives reward for breaking and collecting wood blocks, but has considerable freedom to explore the world. We chose these four environments because they span a range of complexity, freedom, and difficulty, as detailed in Appendix E.

**Agents** The seven agent configurations represented in our dataset include three RL algorithms and two trivial agents for comparison. We selected RL agents spanning the range from extrinsic task reward only to intrinsic motivation reward only. Additionally, we included random and no-op agents, two opposite extremes of trivial behavior. Our goal was to represent a wide range of behaviors: playing to achieve a high score, playing to explore the environment, and taking actions without regard to the environment. Specifically, we used the PPO agent (Schulman et al., 2017) trained to optimize task reward, and the RND (Burda et al., 2018b) and ICM (Pathak et al., 2017) exploration agents using PPO as an optimizer, which can incorporate both an intrinsic reward signal, and an extrinsic reward signal which can be enabled for task-specific behavior, or disabled for task-agnostic behavior. We evaluate RND and ICM in both of these configurations. Each agent is summarized in Appendix F.

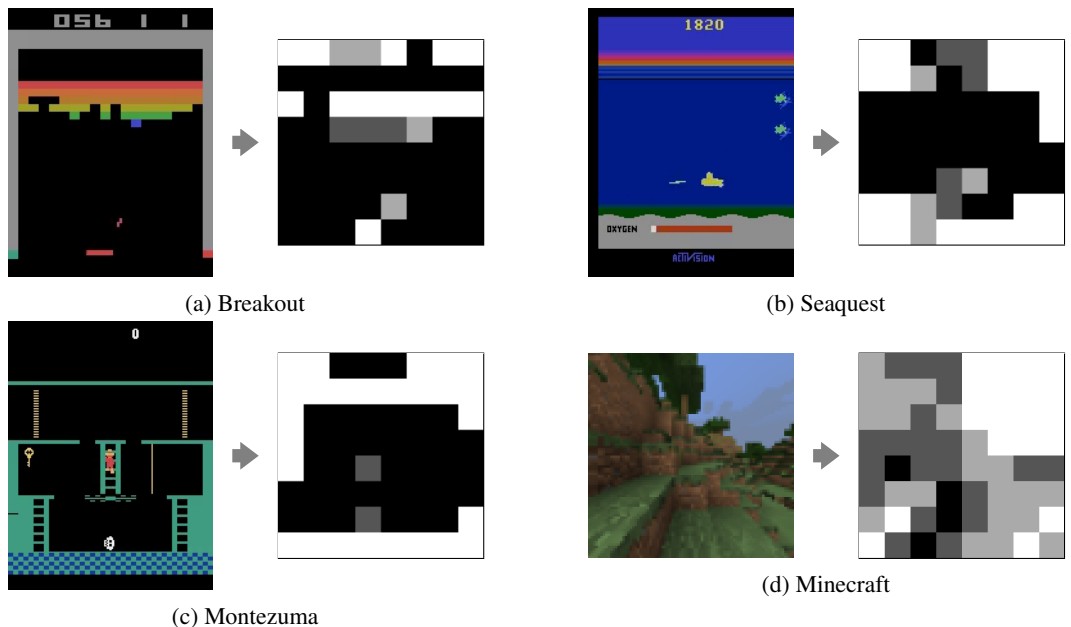

(a) Breakout

(b) Seaquest

(c) Montezuma

(d) Minecraft

Figure 1: Preprocessing used to assign input images to buckets. Similar to Go-Explore (Ecoffet et al., 2019), we resize the images to $8 \times 8$ pixels and discretize each of the resulting cells to one of $4$ values. The examples show that this procedure preserves positions of objects in the game, such as the player, ball, fish, and skull. We enumerate the compactified images to represent each unique frame by an integer index to compute discrete probability tensors for the environments.

## 3 METHOD

The goal of this paper is to evaluate agents using metrics other than task reward. For this, we collected 100 million frames on each of the three Atari environments with each of seven agents: random, no-op, PPO, and RND and ICM with and without task reward. Minecraft was evaluated for 12 million frames per agent because the simulation is slower than the Atari games, and five agents rather than seven were used, excluding both configurations of ICM.

To evaluate the agents, we first preprocessed the data and then computed our metrics in aggregate over the entire lifetime of each agent/environment configuration (yielding one number per metric-agent-environment). In this section, we describe our preprocessing method and introduce our estimators for the five considered metrics.

### 3.1 PREPROCESSING

To efficiently compute the metrics in a robust manner, we discretize the agent's input images so that they can be represented by integer indices. This allows us to summarize each collected datasets as a probability tensor that holds the probability of each transition. Figure 1 visualizes the preprocessing for the three environments.

**Discretization** We first convert the RGB images to grayscale as they were seen by the agents. After that, we bilinearly resize them to $8 \times 8$ pixels. We discretize these low-resolution images to four possible values per pixel, with thresholds chosen as the brightness percentiles 25, 50, and 75 across all unique values of the corresponding pixel in the environment across all agents.

**Aggregation** The unique compactified images are enumerated and summarized into a tensor of counts for each agent and environment combination. For an image index $1 \le i \le |X|$, and action index $1 \le j \le |A|$, and a successor image index $1 \le k \le |X|$, where $X$ is the set of inputs and $A$ the set of actions, the count tensor is defined as,

$$N_{ijk} \doteq \text{number of transitions from image bucket } i \text{ and action } j \text{ to image bucket } k. \quad (1)$$

Normalizing the count tensor $N$ yields a probability tensor $P$ that stores the probability of each transition in the agent's dataset. Under the assumption of a Markovian environment and agent, the probability tensor fully describes the statistics of the preprocessed dataset,

$$P \doteq \frac{1}{\sum_{ijk} N_{ijk}} N, \quad \text{so that} \quad \sum_{ijk} P_{ijk} = 1. \tag{2}$$

The probability tensor $P$ describes the joint probability of transitions for each agent and environment and thus allows us to compute any marginals and conditionals needed to compute the metrics.

## 3.2 METRICS

We compare two task specific metrics, task reward and human similarity, as well as three task-agnostic metrics: input entropy, information gain, and empowerment. The task-agnostic metrics cover the different types of objectives identified by Hafner et al. (2020). We compute a single value of each of these metrics on each agent-environment dataset.

**Task Reward** The reward provided by reinforcement learning environments measures success at a specific task. The environments we use have only one predefined task each, despite the wide range of conceivable objectives in Montezuma's Revenge and Minecraft in particular. This is true of many RL environments, and limits one's ability to analyze the behavior of an agent in a general sense within one environment. While there are multi-task benchmarks, they often include a distinct environment for each task rather than multiple tasks in the same environment (Yu et al., 2019). This would make it difficult to evaluate the agent's ability to globally explore its environment independent of the task.

**Human Similarity** Task reward captures only the agent's success at the specific task defined via the reward function. This may not match up with a human observer's definition of intelligence. We suggest that a more general measure of intelligence may relate to similarity between the agent's behavior and human behavior in the same environment, i.e. using human behavior as a "ground truth". Hence, we propose a human similarity metric that approximates the overlap between the inputs observed by a human player and an RL agent. To compute this, we used the Atari-HEAD dataset (Zhang et al., 2019) and preprocessed the data as described in Section 3.1.

We compute human similarity as the Jaccard index, or intersection over union, between input images encountered in the human dataset and those encountered by the artificial agent. It is simpler than but related to inverse reinforcement learning objectives in the vein of Ziebart et al. (2008); Klein et al. (2012). For this, we first compute the empirical visitation probabilities of inputs from the probability tensors $P^{\text{agent}}$ and $P^{\text{human}}$ of the artificial agent and the human player, respectively. The human similarity is then computed as the fraction of non-zero entries,

$$S \doteq \frac{\left|\left\{i : p(i) > 0 \wedge q(i) > 0\right\}\right|}{\left|\left\{i : p(i) > 0 \vee q(i) > 0\right\}\right|}, \quad \text{where} \quad p(i) \doteq \sum_{jk} P_{ijk}^{\text{agent}}, \quad q(i) \doteq \sum_{jk} P_{ijk}^{\text{human}}. \tag{3}$$

Note that while we use input images from recorded human behavior to compute human similarity, we are not able to compare the human and agent behavior directly, as the RL agents play in an environment with sticky actions, while the humans did not. More generally, human similarity would be challenging to compute in some environments, such as high-dimensional continuous control. Even where it is possible, large human datasets are expensive to collect. Thus, we consider three task-agnostic metrics, which do not require environment-specific engineering or human demonstrators.

**Input entropy** Input entropy measures how improbable the agent's inputs are under a trained density model (Schmidhuber, 1991). It has been used in RL to model an agent's success at exploring its environment, where a higher input entropy score implies a wider variety of states observed, for instance by Oudeyer et al. (2007); Bellemare et al. (2016a); Burda et al. (2018b).

In general, input entropy is the cross entropy of future inputs under a density model trained alongside the agent, but as we retrospectively compute the metric over the agent's lifetime, we only need to compute the probability vector over inputs once. Specifically, we use the marginal entropy over individual inputs, computed by summing the probability tensor over the second and third axes as,

$$C \doteq -\sum_i p(i) \log p(i), \quad \text{where} \quad p(i) \doteq \sum_{jk} P_{ijk}^{\text{agent}}. \tag{4}$$

**Empowerment** Empowerment measures the agent's influence over its environment (Klyubin et al., 2005), which has been applied to reinforcement learning by Mohamed and Rezende (2015); Karl et al. (2017). We use the interpretation of empowerment that measures the agent's realized influence on the world, rather than its potential influence, formalized as the mutual information between actions and sensory inputs (Salge et al., 2014; Hafner et al., 2020).

We compute empowerment mutual information as the difference between the entropy of actions given the preceding input, before and after observing the following input,

$$E \doteq \left( \sum_{ijk} p(i,j,k) \log p(i,j,k) \right) - \left( \sum_{ij} q(i,j) \log q(i,j) \right),$$

$$\text{where} \quad p(i,j,k) \doteq \frac{P_{ijk}}{\sum_{j'} P_{ij'k}}, \quad q(i,j) \doteq \frac{\sum_k P_{ijk}}{\sum_{j'k} P_{ij'k}}. \tag{5}$$

**Information gain** Information gain is a measure of how much the agent learns from its observations Lindley et al. (1956). It is the mutual information between observations and the agent's representation of the environment. Information gain has led to successful exploration in reinforcement learning (Sun et al., 2011; Houthooft et al., 2016; Shyam et al., 2018).

To measure the amount of information gained, we need a way to represent the agent's knowledge about its environment. Preprocessing the agent's inputs into discrete classes enables us to represent its knowledge as a belief over the transition matrix $M$. The total information gain of over agent's lifetime is the entropy difference of its beliefs at the beginning and end of the dataset,

$$I \doteq \mathrm{H}\big[M\big] - \mathrm{H}\big[M \mid \text{dataset}\big] = \mathrm{E}\big[\log p(M \mid \text{dataset}) - \log p(M)\big]. \tag{6}$$

We choose the belief to be a vector of Dirichlet distributions as in (Sun et al., 2011; Friston et al., 2017), each of which has one concentration parameter, initialized at 1, for each possible subsequent input. For each transition that occurs in the dataset, the corresponding concentration parameter is set to 2, which we empirically found to be more effective than varying the parameter based on the number of occurrences. This is explained by the limited amount of stochasticity in the environments even after pre-processing; more than 80% of all state/action pairs transition to only one unique state in Breakout, Seaquest, and Minecraft, so that seeing a transition once establishes that the transition is possible, and seeing it multiple times should not increase the model's confidence by very much.

The entropy of a Dirichlet distribution is given by,

$$\mathrm{H}\big[D\big] = \log \mathrm{B}(\alpha) + \left( \sum_{k=1}^{|X|} \alpha - |X| \right) \psi\left( \sum_{k=1}^{|X|} \alpha \right) - \sum_{k=1}^{|X|} (\alpha_k - 1)\psi(\alpha_k), \tag{7}$$

where $\alpha$ is the vector of concentration parameters, $\psi$ is the digamma function, and B is the incomplete beta function (Lin, 2016). The entropy of the distribution over $M$ is then given by adding up the entropy of each of the Dirichlet distributions.

## 4 ANALYSIS

We conduct a wide range of analyses to understand how the three task-agnostic metrics relate to another and to the supervised metrics of task reward and human similarity. We first compare the agents included in our RL datasets based on their values of the two supervised and three task-agnostic metrics we consider. Next, we analyze correlations between our task-agnostic and supervised metrics. Finally, we discuss correlations between the three task-agnostic metrics themselves. Figure 3 shows correlation matrices of the five metrics, Table 2 contains the tables of all computed metric values, and further visualizations can be found in the supplementary material. We use the OpenAI implementations of ICM (Burda et al., 2018a) and RND (Burda et al., 2018b) and the Stable Baselines implementation of PPO (Hill et al., 2018). We will release the source code for replicating our analyses and the collected datasets and metrics upon publication to spur further work on unsupervised evaluation of RL agents.

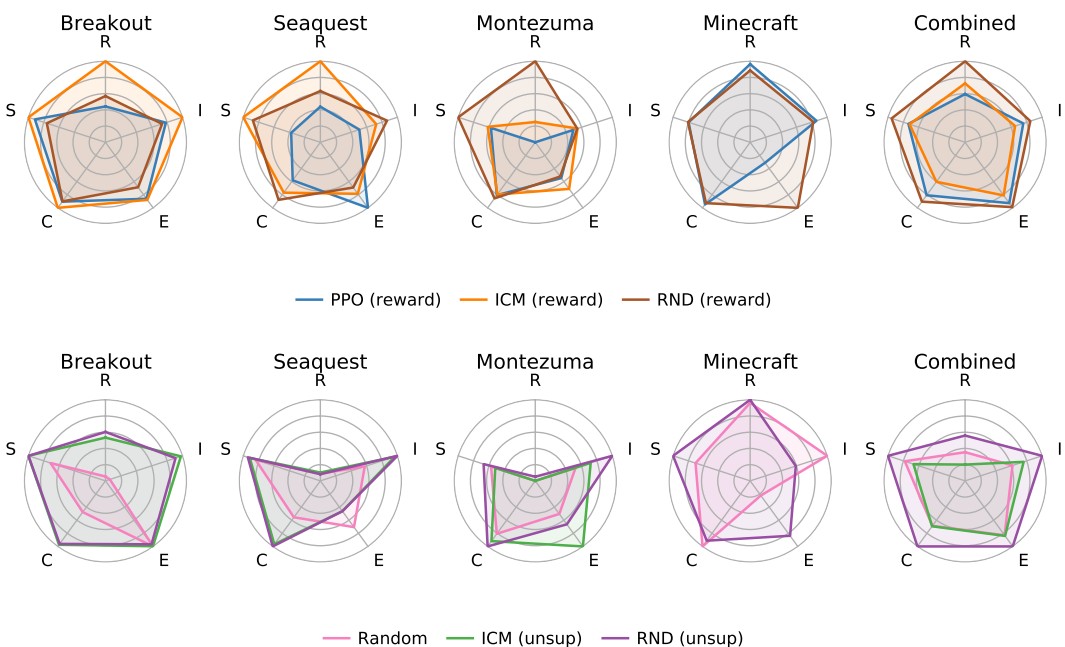

Figure 2: Overview of agent performance under the five metrics. Task Reward (R) and Human Similarity (S) are supervised metrics. Input Entropy (C), Empowerment (E), and Information Gain (I) are task-agnostic metrics. The two task-specific exploration agents achieve the highest task reward and human similarity on average across Atari environments, and RND without reward in Minecraft. ICM or RND each achieve the highest input entropy and infogain value in three out of four environments according to our metrics. More unexpectedly, we find that PPO and task-agnostic ICM achieve high empowerment, even in Montezuma where they achieve low task reward. The no-op agent achieves the lowest scores in all metrics does not show up in the normalized coordinates.

## 4.1 EVALUATION OF AGENTS

**Task reward**   Comparing the mean episode score of task-specific RND and ICM in our agent datasets (Appendix D) with Taïga et al. (2020), we find similar performance of the two agents. Our agents perform better in Breakout, ICM has higher reward than RND in Seaquest which is the reverse in Taïga et al. (2020). We observe in Table 2 that task-specific RND or ICM achieves the highest task reward per time step in all environments, showing that agents benefit from explicit exploration objectives in all three environments.

**Human similarity**   Human similarity is the highest for task-specific ICM and RND in Seaquest and Montezuma respectively, but for task-agnostic ICM in Breakout and task-agnostic RND in Minecraft. We find that exploration agents achieve the highest human similarity in all three environments, as expected.

**Input entropy**   We find that task-agnostic ICM and RND obtain the highest input entropy in all environments, except in Minecraft where the random agent achieves highest input entropy. These results may be related to the fact that ICM and RND maximize input entropy (Appendix F), and would suggest that using extrinsic reward "distracts" the task-specific agents from maximizing input entropy. Minecraft is an outlier among our environments in that random actions can explore a wide range of states by looking around. The no-op agent is always the lowest, and in Breakout and Seaquest all RL agents achieve higher input entropy than the random agent.

**Information gain**   Information gain is highest for the random agent in Breakout, for PPO in Seaquest, for the two configurations of RND in Montezuma, and for the random agent in Minecraft. In all four environments, the agent achieving the highest input entropy also achieves the highest infogain, confirming our observation that input entropy and infogain are related objectives.

**Empowerment**   The agents exhibiting high and low empowerment vary more between environments than for input entropy. We observe that in all three Atari environments, the random agent has relatively high empowerment; higher than all of the RL agents in Breakout (Table 2). This may be related to the

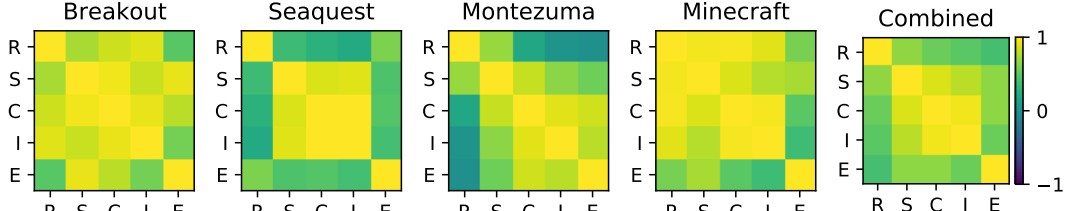

Figure 3: Pearson correlation coefficients between the five lifetime metrics considered in this study: Task Reward (R), Human Similarity (S), Input Entropy (C), Empowerment (E), Information Gain (I). The metrics were computed for the lifetime of each agent and the correlation is taken across agents. Aggregated across environments, all metrics correlate positively. Human similarity correlates substantially with both task reward (0.67) and the task-agnostic metrics (0.89, 0.79, 0.66). Reward only correlates weakly with the task-agnostic metrics (0.54, 0.49, 0.41). As per-environment correlations are over 7 data points in Atari environments and 5 in Minecraft, we do not report them as statistically significant results.

fact that our simple preprocessing method is not semantically meaningful, i.e. it does not provide for generalization across similar but non-identical images. Thus, empowerment may not distinguish well between behavior in which the agent learns over time, and random behavior resulting in many subtly different episodes. This suggests that learning good input representations may key to exploration.

## 4.2 EVALUATION OF TASK-AGNOSTIC METRICS

**Correlation with task reward**   We find that task reward correlates relatively well (0.67) with human similarity. None of the task-agnostic metrics correlate strongly with reward, the highest being input entropy at 0.54. Considering that human similarity itself correlates well with reward, this suggests that input entropy and task reward capture different "components" of agent behavior.

**Correlation with human similarity**   Multiple metrics correlate well with human similarity, the strongest once again being input entropy at 0.89. Human similarity exhibits stronger correlations with the task-agnostic metrics we consider than does task reward. It is worth noting that input entropy correlates the most strongly with both task-specific metrics, as is visually evident in Figure 4. It would be worth considering correlations with human similarity on more open-ended environments in the future, where human players are not necessarily optimizing task reward. This would help determine whether the strong correlations observed with human similarity are a property of the task-agnostic metrics in general, or of the human behavior on Atari games specifically.

**Comparison between environments**   In 23 out of 24 cases in Figure 3, we find that the three task-agnostic metrics correlate positively with task reward and human similarity. The correlations are especially strong in Breakout and Seaquest. Input entropy correlates strongly with reward in Breakout (0.85). Similarly, empowerment is correlated with reward in Seaquest (0.61). We attribute this to the fact that Breakout and Seaquest are reactive games that require fast paced action choices to achieve high task rewards. In Montezuma, the task-agnostic metrics correlate less strongly with reward; in fact, empowerment does not correlate at all (0.00). We suggest this may be related to the very difficult exploration in the game, where the agent is free to take many different courses of action, some of which result in high empowerment, without obtaining any task reward. Minecraft is an open-world environment, but the Treechop task limits the agent to breaking rather than placing blocks, so this specific task is less open-ended. We find that input entropy and infogain correlate very strongly with task reward (0.98, 0.91) and all three task-agnostic metrics correlate with human similarity (0.89, 0.78, 0.74) in Minecraft.

**Combining metrics**   To analyze whether a combination of task-agnostic metrics accounts better for task reward or human similarity than any individual metric, we sum up different task-agnostic metrics. No summation of two or three task-agnostic metrics correlates more strongly with human similarity or task reward than input entropy alone. Additionally, we find that a linear model of input entropy, infogain, and empowerment can predict task reward and human similarity with correlations of 0.55 and 0.91 respectively. This is only slightly better than the correlation with input entropy in the human similarity case (0.89).

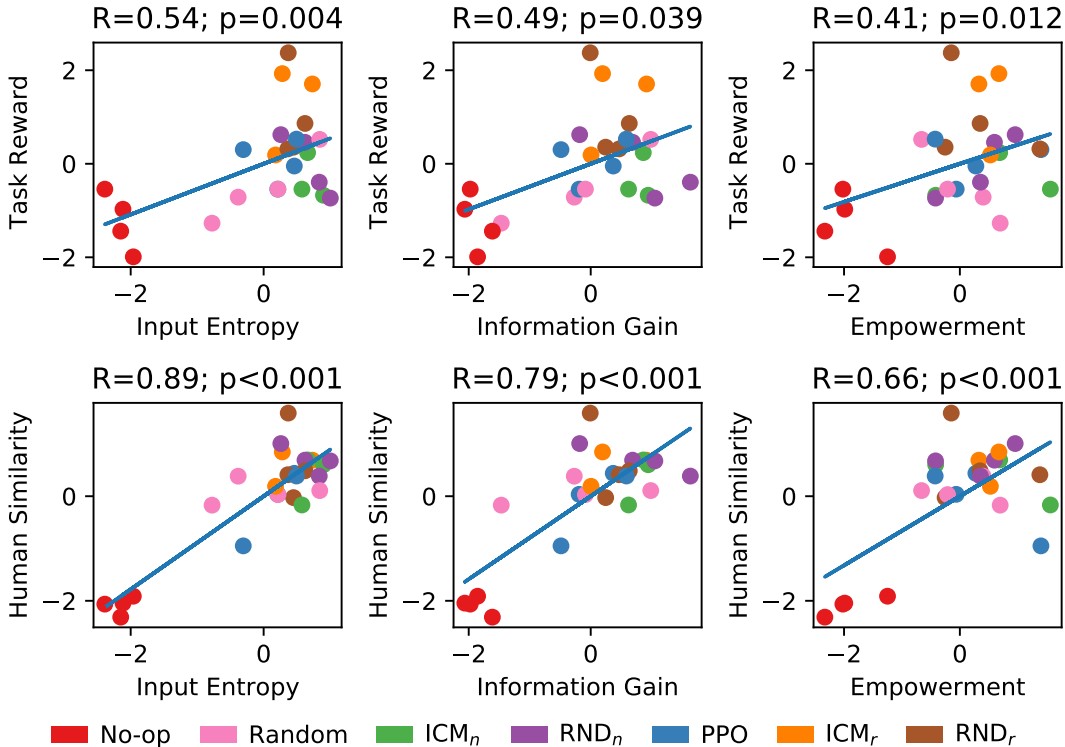

Figure 4: Scatter plots showing correlations between the three task-agnostic metrics (X axis) and the two task-specific metrics (Y axis), normalized for each environment separately. It is visually evident that the task-agnostic metrics show significant correlations with human similarity but are only weakly correlated with task reward. Additionally, note that the no-op configurations are clearly distinguished from the other agents along the input entropy, empowerment, infogain, and human similarity axes, while yielding similar task reward to the random agent and some RL agent configurations. This shows that input entropy, empowerment, and infogain capture aspects of behavior not captured by task reward, in particular distinguishing between the random and no-op agents that behave very qualitatively differently despite achieving similar task reward.

### 4.3 COMPARISON AMONG TASK-AGNOSTIC METRICS

**Correlation among metrics** In Figure 3, input entropy and information gain are shown to correlate strongly with one another (0.95). Input entropy also correlates positively with both of them, though less so (0.66 with input entropy and 0.55 with information gain). This suggests that empowerment explores in a different manner than input entropy and information gain. This finding could guide the design of future exploration methods by suggesting that, while combining metrics does not seem to be beneficial in our environments, empowerment and input entropy/information gain are distinct objectives and so combination of task-agnostic metrics could be applicable in other cases.

## 5 DISCUSSION

In this paper, we have collected large and diverse datasets of agent behavior, computed three task-agnostic metrics on each dataset, and analyzed the correlations of the task-agnostic objectives with task reward and with a human similarity metric. We have found that a simple probability tensor implementation of input entropy shows promise as a task-agnostic objective for RL agent evaluation, and that input entropy and information gain correlate more strongly with human similarity than task reward does.

**Reliability of results** We are confident in proposing input entropy as a task-agnostic exploration metric, given that it correlates the most with task reward and human similarity. We are also confident in our human similarity metric as a supervised baseline. As the human data does not use sticky

actions, we are limited to using a heuristic that does not rely on the action dynamics. Despite this, human similarity correlates strongly with task reward as expected.

**Limitations and future work**

- Our downscaling and discretization method is a simple and transparent preprocessing method, but may not be optimal. More semantically meaningful representations, potentially including deep learning embeddings, may have the potential to uncover additional correlations and/or increase the low overlap observed for our human similarity metric.

- The human dataset we used is limited in quantity compared to agent data (roughly 250K frames per environment, versus 100M frames of agent data in the Atari games). This is a possible reason that our human similarity metric, overlap between agents and human inputs, is low; see Table 2. Access to more human data would be helpful for future work.

- When RL agents are trained to optimize extrinsic task reward, it is clear what task the agent is trying to accomplish. The same is not necessarily true of human data, especially in more open-ended games like Montezuma's Revenge, where players may choose to explore or pursue an objective other than that defined by the reward function. Use of a human similarity metric with respect to distinct human datasets with different tasks for the players could yield some insight on how closely the human concept of exploration aligns with the task in commonly used environments. An example of such a dataset is is the MineRL human dataset (Guss et al., 2019).

**Summary of insights**    Task reward, while a useful measure of agent intelligence, may not be complete. We propose input entropy as a promising task-agnostic metric for agent evaluation, finding that it correlates more strongly with human similarity than does task reward. We also find that input entropy and information gain correlate strongly with each other but to a lesser degree with empowerment, and thus recommend future research into combining empowerment with input entropy or information gain, in a variety of environments.

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

## A  LIFETIME METRICS

**Breakout**

| Symbol | Metric | No-op | Random | $\text{ICM}_n$ | $\text{RND}_n$ | PPO | $\text{ICM}_r$ | $\text{RND}_r$ |
|---|---|---|---|---|---|---|---|---|
| R | Task Reward | 0.0000 | 0.0071 | 0.0695 | 0.0786 | 0.0576 | **0.1302** | 0.0743 |
| S | Human Sim. | 0.0000 | 0.0247 | **0.0346** | 0.0346 | 0.0317 | 0.0346 | 0.0263 |
| C | Input Entropy | 0.0000 | 7.9303 | 16.2240 | 15.9964 | 15.0695 | **16.6389** | 14.9995 |
| I | Infogain | 0.0000 | 0.0203 | 0.3458 | 0.3213 | 0.2770 | **0.3533** | 0.2596 |
| E | Empowerment | 0.0000 | **0.4039** | 0.4028 | 0.3907 | 0.3479 | 0.3548 | 0.2770 |

**Seaquest**

| Symbol | Metric | No-op | Random | $\text{ICM}_n$ | $\text{RND}_n$ | PPO | $\text{ICM}_r$ | $\text{RND}_r$ |
|---|---|---|---|---|---|---|---|---|
| R | Task Reward | 0.0000 | 0.1546 | 0.1789 | 0.1424 | 0.7638 | **1.7379** | 1.0993 |
| S | Human Sim. | 0.0001 | 0.0031 | 0.0034 | 0.0035 | 0.0015 | **0.0037** | 0.0032 |
| C | Input Entropy | 6.7031 | 12.4245 | 16.6600 | **17.0014** | 12.6777 | 14.6161 | 15.7433 |
| I | Infogain | 0.0000 | 0.2209 | 0.3700 | **0.3835** | 0.1941 | 0.2779 | 0.3319 |
| E | Empowerment | 0.0000 | 0.6110 | 0.4013 | 0.4008 | **0.8661** | 0.6806 | 0.5975 |

**Montezuma**

| Symbol | Metric | No-op | Random | $\text{ICM}_n$ | $\text{RND}_n$ | PPO | $\text{ICM}_r$ | $\text{RND}_r$ |
|---|---|---|---|---|---|---|---|---|
| R | Task Reward | 0.0000 | 0.0003 | 0.0000 | 0.2163 | 0.0003 | 1.0620 | **4.2374** |
| S | Human Sim. | 0.0001 | 0.0069 | 0.0063 | 0.0081 | 0.0070 | 0.0075 | **0.0120** |
| C | Input Entropy | 1.9008 | 7.1812 | 7.9123 | **8.4405** | 7.1757 | 7.1073 | 7.4942 |
| I | Infogain | 0.0000 | 0.0120 | 0.0165 | **0.0229** | 0.0113 | 0.0126 | 0.0125 |
| E | Empowerment | 0.0000 | 0.1326 | **0.2629** | 0.1743 | 0.1436 | 0.1869 | 0.1373 |

**Minecraft**

| Symbol | Metric | No-op | Random | $\text{RND}_n$ | PPO | $\text{RND}_r$ |
|---|---|---|---|---|---|---|
| R | Task Reward | 0.0000 | 0.0012 | **0.0013** | 0.0012 | 0.0011 |
| S | Human Sim. | $2.2 \times 10^{-6}$ | $2.8 \times 10^{-5}$ | $\mathbf{4.0 \times 10^{-5}}$ | $3.2 \times 10^{-5}$ | $3.2 \times 10^{-5}$ |
| C | Input Entropy | 9.4112 | **16.2270** | 14.8039 | 15.3758 | 15.0612 |
| I | Infogain | 0.0004 | **0.0583** | 0.0345 | 0.0502 | 0.0477 |
| E | Empowerment | 0.0000 | 0.0770 | 0.2885 | 0.1077 | **0.3444** |

Table 2: Lifetime values of each metric for all agents and environments, with the highest value of each row in bold. In all three environments, the highest task reward is achieved by task-specific RND or ICM, which maximize both task reward and different implementations of input entropy; and the highest input entropy is achieved by task-agnostic RND or ICM, which is to be expected as these agents maximize input entropy alone. Agents with the highest human similarity, empowerment, and information gain vary by environment. Note that the random agent achieves high reward in Minecraft; this may be related to the shorter run-time of 12 million frames, which was necessary because the Minecraft environment is slower than the Atari games. Nonetheless, the random agent in Minecraft has high input entropy and infogain, as expected due to the correlation of those metrics with reward.

## B    INFORMATION GAIN VARIANT CORRELATIONS

| Information Gain Implementation | Shared Discretizations | | Unshared Discretizations | |
|---|---|---|---|---|
| | Task Reward | Human Sim. | Task Reward | Human Sim. |
| Dirichlet of transition counts | $-0.44$ | $-0.36$ | 0.05 | 0.40 |
| Dirichlet of unique transitions | 0.49 | 0.79 | 0.35 | 0.71 |
| Logarithm of transition counts | 0.55 | 0.84 | 0.37 | 0.78 |
| Square root of transition counts | 0.52 | 0.84 | 0.38 | 0.78 |

Table 3: Correlations of four information gain implementations with task reward and human similarity. We compare the implementations for two forms of preprocessing, which use the same discretization across agents or use a different discretization for each agent based on its data only. We find that the Dirichlet distribution of unique transitions and logarithm and square root of transition counts exhibit greater correlations when using shared discretizations, and that all of the said three methods correlate strongly with human similarity and more weakly with task reward.

## C    HUMAN SIMILARITY VARIANT CORRELATIONS

| Human Sim. Implementation | Task Reward | Input Entropy | Information Gain | Empowerment |
|---|---|---|---|---|
| Jaccard Similarity | 0.67 | 0.89 | 0.79 | 0.66 |
| Jensen-Shannon Divergence | 0.26 | 0.77 | 0.67 | 0.66 |

Table 4: Correlations of two human similarity implementations with task reward and the three task-agnostic metrics. We compare the Jaccard similarity (intersection over union) of the set of states visited by the human player and those visited by the RL agent, with the Jensen-Shannon divergence between the two sets. We find that Jaccard similarity correlates much more strongly with task reward, slightly more strongly with input entropy and information gain, and near-equally with empowerment, as compared to Jensen-Shannon divergence. The two implementations have a correlation of 0.78 with each other.

## D    EPISODE RETURNS

| | Naive | | Task-Agnostic | | | Task-Specific | |
|---|---|---|---|---|---|---|---|
| | No-op | Random | ICM | RND | PPO | ICM | RND |
| Breakout | 0.0000 | 1.6513 | 92.9397 | 132.7004 | 105.6570 | **240.3896** | 147.6850 |
| Seaquest | 0.0000 | 98.9152 | 330.5743 | 349.1177 | 1713.8523 | **4493.0141** | 2700.6128 |
| Montezuma | 0.0000 | 0.6422 | 0.0130 | 489.0153 | 0.7339 | 238.7989 | **5186.5680** |

Table 5: For our analysis we normalized all metrics by the number of time steps in the agent's dataset. For comparison with prior work, this table shows the unnormalized episode returns of all agents. These verify that the agents were trained correctly. Note that they each correspond to only one random seed.

# E ENVIRONMENTS

We consider three Atari environments: Breakout, Seaquest, and Montezuma's Revenge. All three games are 2D environments with backgrounds fixed relative to the screen. The player is free to move around within the screen on one axis in Breakout, and two axes in the Seaquest and Montezuma. In Montezuma, the player can additionally navigate from one room into another.

**Breakout**   Breakout is a game in which the agent controls a paddle at the bottom of the screen with the objective of bouncing a ball between the paddle and the blocks above, which disappear upon contact with the ball. The game is nearly deterministic: the only source of randomness other than sticky actions is the initial direction of the ball after starting the game or losing a life. Breakout is the simplest of our three environments, the player only being free to move a paddle in one dimension.

**Seaquest**   Seaquest is a game in which the player controls a submarine, with the objective of defending oneself against sharks and other submarines which appear frequently and randomly at both sides of the screen. Additionally, the agent is tasked with picking up divers, which also appear at random, and bringing them to the top of the screen. Because of the random appearance of sharks and divers, the game can be difficult to predict, and made more so by sticky actions. It is more challenging that Breakout, as the player moves along two dimensions and enemies appear at random; however, the agent's task is reactive, with no long-term planning required.

**Montezuma**   Montezuma's Revenge is a difficult platformer game with a large first level consisting of many rooms, which necessitates long-term planning. The player must navigate ladders, ropes, and various hazards such as moving skulls and lasers. Rewards are very sparse, and given only when the player completes an objective such as finding a key or opening a door, which often require complex and specific action sequences. For this reason, intrinsic rewards (Burda et al., 2018b) or human demonstrations (Aytar et al., 2018) are important to succeed at the game.

**Minecraft Treechop**   MineRL (Guss et al., 2019) is a set of environments in Minecraft, a block-based 3D game in which the player can explore, build, and mine within a procedurally generated world. The Treechop environment provides the player with an axe and restricts the action space such that the player can walk around and break blocks, being given task reward for breaking and collecting wood blocks from trees. Though the goal is clearly defined, there are a wide range of possible activities the agent can pursue.

We follow the standard evaluation protocol (Machado et al., 2018). The Atari games yield $210 \times 160 \times 3$ images, which are converted to grayscale and rescaled to $84 \times 84$ before being input to the agent; MineRL returns $64 \times 64$ images which are input to the agent directly. The agent chooses one of a set of discrete actions: 4 in Breakout, and 18 in Seaquest and Montezuma. While the effects of the actions in the original games are deterministic, we use "sticky actions", meaning that the agent's action is ignored with $25\%$ chance and instead the previous action is repeated.

## F  AGENTS

**Random**    An agent that uniformly samples random actions from the available action space.

**No-op**    The three environments we consider have a no-op action which does nothing, though the environment still continues to update. We consider an agent which always takes this no-op action.

**PPO**    Proximal Policy Optimization (Schulman et al., 2017) trained on extrinsic rewards only. PPO is a commonly used policy gradient algorithm that optimizes the task reward on-policy. It optimizes task reward while preventing the policy from changing too much on each training step, so that the learning process is stable.

**ICM**    Intrinsic Curiosity Module (Pathak et al., 2017) is an exploration agent that maximizes input entropy in addition to the task reward signal. Image embeddings are created that incorporate only those aspects of the image that can affect the agent, by training a network to predict the agent's action given the preceding and following input images. A second network is trained to predict an input image embedding given the preceding input and action. Its error is used as an intrinsic objective.

**RND**    Random Network Distillation (Burda et al., 2018b) is an agent based on PPO that maximizes input entropy, implemented as the prediction error of a model trained to distill a random function. It makes use of a randomly initialized and fixed neural network which predicts an embedding from the agent's input images. Another network is trained on the outputs of the random network, and its error, which models the agent's "familiarity" with the inputs in question, are used as an intrinsic objective.

