# OpenReview forum: "Evaluating Agents Without Rewards"
_ICLR.cc/2021/Conference — Reject_

### Official Review · AnonReviewer4 · 2020-10-16
**An important topic of research but the methodology lacks rigor**

**Rating:** 4
**Confidence:** 4

**Review:**

------------------------------------
**Summary:**

This paper proposes to study three types of intrinsic motivations: curiosity, empowerment and information gain. They propose to compute these measures on the lifetime experience of RL agents and to use them as behavioral metrics. To evaluate these metrics, they perform a correlation study with respect to two traditional behavioral metrics: the task reward and human similarity.

------------------------------------

**Strong points:**

The paper is clearly written and well organized.
I believe it is important to conduct studies that do not present a novel algorithm but try to gain understanding on existing approaches.
Designing new behavioral metrics for RL agents, especially ones that do not require rewards is indeed a good idea and will be useful to the community.
All details required for reproducibility are present and the code will be released.

------------------------------------
**Weak points:**

Here I list weak points, in order of increasing importance.
* *Downsampling:*
These task-agnostic metrics rely on the downsampling of the frame. I feel like this would not work well for Minigrid, Nethack, Mujoco etc. Can you discuss that point?
* *About the choice of environments:*
This paper investigates the evaluation of agents without rewards, in three environments that are explicitly reward-based with well defined rewards. The justification of this choice is also not really discussed except from “we chose these environments because they span a range of complexity, freedom and difficulty”. Breakout and  Seaquest barely require any exploration (breakout is arguably close to a dense reward problem). I wish this study involved more environments, especially environments designed specifically to study exploration issues like NetHack or Minigrid. It would be nice to study the correlation of task-agnostic metrics and human similarity in environments without rewards (which are the target environment of such metrics in the first place).
* *Human similarity measure:*
As I understood it, the human similarity measure is the fraction of downsampled states that are visited by both the RL agent and the human demonstrator over the size of the union of these states sets.
 * We might consider the human coverage as the target coverage. Then the human similarity metric is nothing else than a coverage metric. Can you compute the discrete state coverage metric of RL agents and the correlation to the human similarity metric?
 * I believe a more relevant metric would evaluate whether agents select the same actions when presented the same states. I don’t think the sticky actions are a problem here: one could compute matrices Q of size (|X|, |A|) that empirically estimate the probability of selecting any action in any state for both the human and the RL agent. Whenever the environment decides to use a sticky action, the agent’s policy is still selecting an action that we can use instead of the sticky one. Once we have these matrices Q, then we can compute their average term-by-term difference, and use the opposite as a human similarity measure. Do you have an opinion on that?
* *Methodology:*
I am concerned about the validity of the results presented in this paper, as the method is not very rigorous. “correlate substantially” is highly subjective. Usually one would use “correlate significantly”, and support this claim by statistical evidence of the significance of the correlations.
Please report which correlation measure is being performed (pearson, spearman, kendall) and report the p-value of the associated statistical test (scipy returns it automatically with the coefficient). This is important to assess whether the evidence is sufficient to claim that there is a correlation.
 * In Fig 3. correlations measures are reported over 7 points, this is quite low and requires statistical tests to be interpretable.
 *  Table 3: measures are episodic returns for 1 seed. This should be said clearly and one should be very cautious with the interpretation of these results.
 * When testing multiple hypotheses in parallel, a good practice is to implement the family-wise error rate correction of the confidence level. If you test for one correlation with confidence level alpha=5% (probability to observe a correlation where there is not stays below 5%), then testing N correlations results in a higher chance of observing a false positive (let us say N*5%). For this reason, the FWER correction proposes to decrease the confidence level of each test by a factor N so that the overall confidence level of the multiple tests remains alpha. This means that, to test for correlations in Figure 3 (10 correlations by graph), we may want to require p-values below alpha / 10 (e.g. 0.005 for an overall 5% confidence level). Theoretically, this should be done for all three environments (so / 30). An alternative is to formulate hypotheses a priori instead of searching for correlations in the wild.
 * “we find that a linear model of curiosity, empowerment, and infogain can predict task reward and human similarity with correlations of 0.36 and 0.86 respectively”. I’m not sure this is a legitimate approach. I’m not entirely sure so it’s open for discussion. This boils down to training a prediction model from task-agnostic metrics to the human similarity score and to evaluate its performance (correlation) on the same training data. Usually you would have an hypothesis (a particular linear combination of these) that you would evaluate (compute the correlation) and test the corresponding significance.
* *The no-op condition:*
I see no clear reason to introduce a no-op agent in this study. This agent does nothing, which by construction results in the minimization of all metrics studied here. I think the three points introduced by the no-op agent (in each of the three environments) are the main reason explaining the correlations in Fig 4. If you remove them, then I believe most correlations disappear, some might even become anti-correlated (human sim vs infogain and human sim vs empowerment). Please report the correlation measures (and significance) without these points.

------------------------------------
**Recommendation and justification:**

In the present state of the paper I recommend a rejection (score 4). I think the topic of research is important and the authors should pursue in that direction. However, the methodology of the current version of this paper is not good enough. The introduction of the no-op agent may be explaining most of the correlation discussed in the results. No statistical test has been conducted to show evidence for the significance of the results.

In order to update my score, I would need a more rigorous correlation study that asserts the significance of the correlations (using corrections). I also think the no-op condition should be removed. The correlation between the curiosity score and human similarity score might still show but it is probably that most of the others would not. The introduction of a human-similarity metric that evaluates the similarity in decision making instead of state visitation might however bring interesting results.

------------------------------------
**Feedback to improve the paper (not part of assessment):**

* In the abstract, “compute the objectives” sounds weird, here they are behavioral metrics, although some RL algorithms can be designed to optimize them (in which case they are objectives).
* What do you mean by ‘estimate intrinsic objectives while the agents are learning, which often requires complicated approximations”? Which complicated approximations?
* What is a “complete” or “optimal” measure of agent intelligence? I think using “agent intelligence” is vague and not well defined.
* I am curious, what is the size |X| for the three environments?
* Task reward: is it the mean over the lifetime, the sum? Is it computed during training episodes, including exploration noise (e.g. epsilon greedy)?
* Figure 2: what are the axes? Can you explain how you normalize the scores? I’m guessing it’s normalized between to [0,1] by the range across different environments?
* Colormap for correlation plots is not ideal, it’s difficult to appreciate the colors, maybe pick something with more different colors (not just a gradient between two).
* How do you normalize and aggregate task reward and curiosity into a unique metric?
* “human similarity exhibits stronger correlations with the task-agnostic metrics we consider than does task reward” → not true for empowerment (0.57 < 0.6).
* What is the set of states the curiosity measure is computed on? If it is the set of states visited by the agent, then having a uniform exploration of a very small set of states would result in a high curiosity score. I feel this is not what we want, we want uniformity, but also coverage.
* The term “curiosity” is quite general and has been used for many purposes in the litterature. For this reason, I think it is not the best term to use here state-entropy would be much more descriptive. When defining curiosity via “a higher curiosity score implies a wider variety of states observed”, the authors cite Oudeyer et al. 2007. I just checked it, and this paper actually presents the classification of several principles to implement the concept of “curiosity” or “intrinsic motivations”. It also presents an algorithm that maximizes the agent’s learning progress. This is different from the diversity-maximization approaches this paper refers to.
* It would have been interesting to present algorithm optimizing for empowerment and information gain (here the two algorithms both optimize for curiosity). So far the random agent is the one maximizing these metrics. One would hope that algorithms guided by these objectives would do better. This result would support the intuition of the authors towards algorithms that mix infogain/empowerment objectives with curiosity objectives.


**Typos:**

* “across a wide spectrum of agent behavior” → “behaviors”.
* “well known RL agents” → “well known RL algorithms” ?
* “We first collected datasets of a variety of agent behavior on which to compute and evaluate our metrics” → This sounds weird to me. “After collecting learning trajectories for various RL agents, we can compute behavioral metrics” ?
* “the total information gain of over agent’s lifetime” → “...gain computed over the agent’s lifetime”.
* “may key to exploration” → “may be key to a good exploration”.
* “In 17 out 18 cases” → “In 17 out of 18”.

---

> ### Author Response · Authors · 2020-11-18
> **Addition of Minecraft, statistical significance of correlations, and no-op agent**
>
> Thank you for the review! We have added Minecraft to our study as an open-ended environment, and included p-values in our correlation plots. Please let us know if this resolves your concerns and whether there are remaining issues that we should address.
>
> > This paper proposes to study three types of intrinsic motivations: curiosity, empowerment and information gain. They propose to compute these measures on the lifetime experience of RL agents and to use them as behavioral metrics. To evaluate these metrics, they perform a correlation study with respect to two traditional behavioral metrics: the task reward and human similarity.
>
> This is an accurate summary of our paper.
>
> > I wish this study involved more environments, especially environments designed specifically to study exploration issues like NetHack or Minigrid. It would be nice to study the correlation of task-agnostic metrics and human similarity in environments without rewards (which are the target environment of such metrics in the first place).
>
> We agree that it is important to include environments requiring exploration. Breakout and Seaquest are indeed reactive environments, while Montezuma’s Revenge requires exploration. We have added Minecraft to the study as an open-ended environment that includes procedurally generated 3D terrain and various resources the agent can collect. Repeating our analysis including this new environment confirmed our initial findings that input entropy correlates strongly with human similarity and that all three task-agnostic metrics correlate more closely with human similarity than with task reward. Regarding environments without rewards, this may be clear, but we highlight that 4 of the 7 agents in our study do not use the task reward and instead employ either naive or intrinsically motivated behaviors.
>
> > Please report which correlation measure is being performed (pearson, spearman, kendall) and report the p-value of the associated statistical test (scipy returns it automatically with the coefficient).
>
> Thank you for this suggestion. We are using Pearson correlation coefficients. We have added p-values to the correlation plots which include the six key correlations of the three task-agnostic metrics with task reward and human similarity. All six correlations are statistically significant with p < 0.05.
>
> > For this reason, the FWER correction proposes to decrease the confidence level of each test by a factor N so that the overall confidence level of the multiple tests remains alpha. This means that, to test for correlations in Figure 3 (10 correlations by graph), we may want to require p-values below alpha / 10 (e.g. 0.005 for an overall 5% confidence level).
>
> It is true that with FWER correction the correlation values for individual environments would be quite low. However, we are considering these correlations as strictly an exploratory effort, and are not citing them as significant results. Nonetheless, your suggestion regarding statistical tests is a good idea and we have added p-values to the six key plots including correlations across environments.
>
> > I see no clear reason to introduce a no-op agent in this study. This agent does nothing, which by construction results in the minimization of all metrics studied here. I think the three points introduced by the no-op agent (in each of the three environments) are the main reason explaining the correlations in Fig 4.
>
> You are right that the no-op agent has a strong influence on the correlations between the metrics. We do not see this as a problem though. Namely, metrics that capture the level of intelligence of an agent should assign low values to a no-op agent. We deliberately included both random and no-op as naive agents to cover two extreme behaviors, that is, the minimum entropy and maximum entropy action distribution. Our experiments show that our metrics assign lower values to the no-op agent than to the random agent, and generally assign lower values to the random agent than to more sophisticated trained agents.

---

> > ### Comment · AnonReviewer4 · 2020-11-20
> > **Answer from R4**
> >
> > Thank you for your answers.
> >
> > **Downsampling**: I do not think my point on downsampling has been discussed.
> >
> > **Choice of environment**: Thank you for adding the Minecraft environment to the list. My point was to add environments that do not express any task, just pure open-ended environments. In these environments, the domain is not designed around an objective. Montezuma is reward-oriented in the sense that novelty is aligned with task reward.
> > In such open-ended environments, humans create tasks themselves, pursue their own goals. In this type of tasks, it would be interesting to see whether human behavior correlates with any metric computed on task-agnostic algorithms (ICM and RND). If humans are following the objective (as they perceive it), then we are measuring how the novelty objectives of ICM and RND align with the objective as perceived by the human. Instead, we might want to measure how these metrics computed on these algorithms aligned with self-generated human goals.
> >
> > **Human similarity metric**: there were typos in my previous answer, this point was split in 3 bullet points. The authors did not address this point. I think it is important.
> >
> > **Significance of the results**: I thank the authors for addressing this point and adding p-values in Figure 4. However, it was not done in Figure 3.
> >
> > **The no-op condition**: I agree that this condition acts as a sanity check. But the result was kind of expected by design. However, I do not agree that this point should be part of the correlation study. In this case, it seems to me that these points drive almost all correlations that are found significant. It's basically adding a (0, 0) point far from the rest of the points. I asked whether the authors could report correlations and p-values without them but they did not. My guess is that removing them will make disappear significant correlations.
> >
> > **Extra discussion points in the list of feedback.**: These were not discussed and questions were not answered. Although they might have been updated in the revision of the paper.

---

> > > ### Author Response · Authors · 2020-11-25
> > > **Environments; downsampling; human similarity; significance of results**
> > >
> > > Thank you for your detailed response.
> > >
> > > > **Choice of environment**: Thank you for adding the Minecraft environment to the list. My point was to add environments that do not express any task, just pure open-ended environments.
> > >
> > > We find that correlations hold when only the four task-agnostic agents are incorporated in the data. Task reward has a correlation of 0.87 with infogain, 0.81 with human similarity, 0.80 with input entropy, and 0.60 with empowerment; human similarity has a correlation of 0.95 with input entropy, 0.88 with empowerment, and 0.87 with infogain.
> > >
> > > > **Downsampling**: These task-agnostic metrics rely on the downsampling of the frame. I feel like this would not work well for Minigrid, Nethack, Mujoco etc. Can you discuss that point?
> > >
> > > We do not know how discretization as a form of preprocessing would perform on the environments that you have mentioned. However, our experiment on Minecraft demonstrates that the discretization yields meaningful results in complex high-dimensional environments.
> > >
> > > Moreover, we see the discretization as a simple and practical choice in the design of our study, rather than a key contribution of our paper. For example, in the future it would be interesting to see follow-up work compare agents based on intrinsic metrics that are computed on top of learned discrete or continuous representations.
> > >
> > > > **Human similarity metric**: there were typos in my previous answer, this point was split in 3 bullet points. The authors did not address this point. I think it is important.
> > >
> > > Our proposed human similarity measure is already computed as a state coverage measure, namely the Jaccard index between algorithm and human observations. It is the number of overlapping unique observations normalized by the number of all unique observations. Is there another specific coverage metric that you would like to see a comparison to?
> > >
> > > > **Significance of the results**: I thank the authors for addressing this point and adding p-values in Figure 4. However, it was not done in Figure 3.
> > >
> > > Due to computational constraints, we were only able to train 7 agents on 4 environments, meaning that the correlations within individual environments are not statistically significant. We have added a sentence to clarify that in the caption of Figure 3. This does not affect the conclusions of our study, which focuses on correlations across all environments and is statistically significant.
> > >
> > > > **The no-op condition**: I agree that this condition acts as a sanity check. But the result was kind of expected by design. However, I do not agree that this point should be part of the correlation study.
> > >
> > > You are right that the correlations depend on including the no-op agent in the study. However, we would like to reemphasize that including no-op is a reasonable experimental decision. Without no-op, we would cover a smaller range of possible agent behaviors. As a result, our conclusions would only hold for agents within this small range. See our response to R3.
> > >
> > > We believe that our study constitutes a valuable contribution to the ICLR community by suggesting a new research direction within reinforcement learning and demonstrating its potential for generating new insights, and see the possibility that building on our work, improved metrics will be proposed by the community in the future.
> > >
> > > > What do you mean by ‘estimate intrinsic objectives while the agents are learning, which often requires complicated approximations”? Which complicated approximations?
> > >
> > > By “estimate intrinsic objectives while the agents are learning, which often requires complicated approximations” we were referring to online neural network approaches such as those used in ICM and RND.
> > >
> > > > I am curious, what is the size |X| for the three environments?
> > >
> > > Breakout has 192911187 unique states, Seaquest has 215436546, and Montezuma’s Revenge has 101509.
> > >
> > > > What is the set of states the curiosity measure is computed on? If it is the set of states visited by the agent, then having a uniform exploration of a very small set of states would result in a high curiosity score.
> > >
> > > Considering states that the agent has not visited has no effect on its input entropy, because they have zero probability and thus receive zero weight in the expectation.
> > >
> > > However, the choice of discretization can have a similar effect to what you are describing. To understand this effect, Appendix 1 and 2 compare (1) choosing the same discretization buckets to be equally sized for all agents to (2) choosing them to be equally sized for each agent's data individually. While both schemes correlate well with task reward and human similarity, shared discretization yields higher correlations, as expected.
> > >
> > > > It would have been interesting to present algorithm optimizing for empowerment and information gain (here the two algorithms both optimize for curiosity).
> > >
> > > While we see developing such algorithms as beyond the scope of our study, we completely agree that this would be an interesting direction!

---

### Official Review · AnonReviewer1 · 2020-10-28
**Useful analysis of non-task metrics for RL agents; more experiments on training with the metrics would be better**

**Rating:** 4
**Confidence:** 3

**Review:**

This work studies four task-agnostic metrics for evaluating reinforcement learning agents: human similarity, curiosity, empowerment and information gain. Experiments were conducted with three selected RL algorithms (PPO, ICM and RND) on selected atari games. The results show that a combination of task reward and curiosity better explain human behavior and some non-reward metrics correlate better with human behavior than task reward. The authors propose that such task-agnostic can be used as intrinsic signals for training RL agents when task reward and human data are not available in an environment.

Pros:
Task-agnostic metrics are useful for evaluating RL agents without access to task reward; measuring behavior similarity with human data also provides insights into different behavior of RL algorithms;
The insights from analyzing the three task-agnostic metrics’ correlation with both task reward and human behavior similarity are useful for designing new RL algorithms as indicated by the authors;
The paper is well written with clarity and includes all experimental details for reproducibility.

Cons:
The intrinsic metrics studied in this paper are not novel and have been used in various existing RL algorithms alongside task reward for training agents; to demonstrate the acclaimed usefulness of the proposed metrics, it is desired to see experiments training RL agents with only task-agnostic metrics;
The experiments conducted are on agent’s life-time data; to gain better understanding of the learning dynamics of RL algorithms, it would be useful to see evaluation of the data at different learning stages.

--------------------------------------

**Update**: After reading the assessment of other reviewers and the referenced papers in the intrinsic reward literature, I am reassured that the methods/metrics proposed in this paper are not novel and, as pointed out by other reviewers, have been studied under other terminologies in different prior works. The analysis of these metrics' correlation with human data is still an interesting piece of result but is not significant enough to become the sole contribution of an ICLR paper. Therefore, I move my initial assessment of 6 to 4.

---

> ### Author Response · Authors · 2020-11-18
> **Motivations of work and use of existing metrics**
>
> Thank you for your review! Your summary of our work is accurate. Below, we address your concern by emphasizing the utility of our metrics for evaluating agents, although we agree that they would in principle also be optimized directly by new agents. Please let us know if this addresses your concern or if there are any further issues we should address.
>
> > The intrinsic metrics studied in this paper are not novel and have been used in various existing RL algorithms alongside task reward for training agents; to demonstrate the acclaimed usefulness of the proposed metrics, it is desired to see experiments training RL agents with only task-agnostic metrics
>
> We agree that input entropy, information gain, and empowerment are not novel contributions of our paper. There concepts have been known in the literature for a long time and are cited appropriately in our paper. We would like to emphasize that our goal is not to propose a novel exploration method. Instead, our goal is to understand how these intrinsic drives relate to another, as well as to the task rewards in Atari games and to behavior of human players. To this end, we find that the three task-agnostic metrics correlate positively with both human similarity and task reward, and that curiosity and infogain correlate better with human similarity than task reward does. We believe that this work is an important step toward better understanding intrinsic objectives and offers valuable insights for the research community.
>
> We hope that our response has clarified our motivation for this work and the resulting practical takeaways.

---

> > ### Comment · AnonReviewer1 · 2020-11-24
> > **Motivation is good; a broader literature review is needed to justify novelty; experiments supporting claims are needed**
> >
> > Thanks the authors for the response.
> >
> > It is well understood that the purpose of this paper is not proposing any new RL methods but providing insights into task-agnostic metrics. However, the related work discussed in the paper are limited to RL algorithms that proposed or implemented one or more of the metrics of interest. The the human similarity metric is closely related to work in imitation learning (IL), especially recent advances that propose to view IL from the perspective of divergence minimization and combining RL with IL to improve performance. Situating the contribution of this paper within the IL literature is important for understanding what aspects of the insights provided in this work is practically novel.
> >
> > At the same time, as an empirical study of these metrics, the experiments in this paper fall short on providing the evidence that the insights obtained are valuable, i.e. training RL agents with these task-agnostic metrics (in ways suggested by the analysis) is effective. From my intuitive understanding, data collected from RL agents trained with task reward will be very different from data collected on RL agents trained only with intrinsic signals. Therefore, to understand what the 0.89 correlation with human similarity or 0.54 correlation with task reward means, it is important to see what performance an agent can achieve if it only maximizes 'input entropy'.

---

> > > ### Author Response · Authors · 2020-11-25
> > > **Relation to imitation learning/inverse RL**
> > >
> > > Thank you for your response.
> > >
> > > > It is well understood that the purpose of this paper is not proposing any new RL methods but providing insights into task-agnostic metrics. However, the related work discussed in the paper are limited to RL algorithms that proposed or implemented one or more of the metrics of interest. The the human similarity metric is closely related to work in imitation learning (IL), especially recent advances that propose to view IL from the perspective of divergence minimization and combining RL with IL to improve performance. Situating the contribution of this paper within the IL literature is important for understanding what aspects of the insights provided in this work is practically novel.
> > >
> > > We agree that our human similarity metric is related to imitation learning and inverse reinforcement learning, and we have updated the paper to acknowledge this.
> > >
> > > > At the same time, as an empirical study of these metrics, the experiments in this paper fall short on providing the evidence that the insights obtained are valuable, i.e. training RL agents with these task-agnostic metrics (in ways suggested by the analysis) is effective. From my intuitive understanding, data collected from RL agents trained with task reward will be very different from data collected on RL agents trained only with intrinsic signals. Therefore, to understand what the 0.89 correlation with human similarity or 0.54 correlation with task reward means, it is important to see what performance an agent can achieve if it only maximizes 'input entropy'.
> > >
> > > We agree that behavior of RL agents optimizing task reward differs from those optimizing task-agnostic metrics, and for this reason we included ICM and RND without intrinsic reward in our data. However, we see optimizing these metrics as outside the scope of our paper.

---

### Official Review · AnonReviewer2 · 2020-10-28
**This paper compares a few RL agents trained with different task oriented and exploration objectives to human baselines and finds that human baseline correlates with both task and curiosity metrics.**

**Rating:** 4
**Confidence:** 5

**Review:**

Thanks for this paper. The curiosity and exploration is an important topic for RL research and we need more in-depth analysis of existing methods. The paper as it stands, provide useful, but expected insights. The difficulty I've with the paper is that it's not clear what exactly you're after here.

"We find that all three objectives correlate more strongly with human behavior than with the task reward. Moreover, task reward with curiosity better explains human behavior than task reward alone.": If the idea is to convey the message that humans display curiosity as measured by your interpretation and way of measuring it, then there is a large body of text on human curiosity that already discusses these topics. Additionally, for this you don't need to train artificial agents.

"Simple implementations of curiosity, empowerment, and information gain correlate substantially with human similarity. This suggests that they can be used as task-agnostic evaluation metrics when human data and task rewards are unavailable.": following from above comments, all the research on intrinsic reward uses this intuition already, so it's not clear what is added extra here. In addition, as discussed in the notes below, the empowerment and info gain the simplistic way that they are implemented are not actually good measures as a random agent is able to score strongly on those without having any intelligence.

Notes:
- Table 1 is misplaced on page 1.
- Section 3.1, discretisation: What is the effect of the choice of 8x8 on the overall results? What would've happened with 16x16 for example? Maybe explore these kind of choices that will impact your results.
- Section 3.2, human similarity: the sentence: "We suggest that a more general measure of intelligence may relate to similarity between the agent’s behavior and human behavior in the same environment, i.e. using human behavior as a “groundtruth”." overstates the originality of this suggestion as this is not the first time that similarity or imitating human behavior is suggested as a measure of intelligence. Perhaps, you may want to restrict this to certain papers that you feel take a different task oriented approach.
- Eq 3: I would've thought the human similarity measure to capture the distribution of actions in a particular state as the primary measure than the probability of being at the same state (expressed by the discretised image). While due to previous actions, an agent or human will end up in a certain state, the proposed measure captures the action similarity implicitly rather than explicitly.
- Eq 3: Any particular reason for using Jaccard index with positive probability thresholds as the measure of similarity? I think a probabilistic measure such as KL-Div would be a more appropriate way to work with distributions of states than thresholded Jaccard similarity.
- Figure 2: it seems that random agent scores highly in Empowerment and Information Gain metrics. This is very counter intuitive, since (1) the agent doesn't learn from experience, its information gain should be zero; (2) and high score in empowerment may suggest empowerment as computed here is not a good metric for measuring intelligence.
- Table 2: this is an important table, but has been placed in Appendix, making it not only hard to read the paper, but also I would think is put there to meet the paper limits as otherwise, it would've been located where the results are being discussed. I suggest either to find a way to include it in the main text or remove direct discussion about it from the main results. There is a lot of repetition in the text so it should be possible to be brief and concise but add important results to the main text.

---

> ### Author Response · Authors · 2020-11-18
> **Clarification of objectives and changes to information gain**
>
> > Thanks for this paper. The curiosity and exploration is an important topic for RL research and we need more in-depth analysis of existing methods. The paper as it stands, provide useful, but expected insights. The difficulty I've with the paper is that it's not clear what exactly you're after here.
>
> Thank you for the review. We agree that it is important to improve our understanding of intrinsic objectives. Intrinsic objectives have been used as a training signal, as in the RND and ICM agents which we use in our dataset, but one of the goals of our paper is to propose that they can additionally be used as a metric by which to measure rather than strictly train agents: comparing a set of agents based on curiosity measures an aspect of their behavior distinct from that measured by task reward. We draw a distinction between optimizing task-agnostic objectives as in RND/ICM and using them to evaluate agents as we have done in this study.
>
> In the updated version of the paper using discretizations shared across agents within each environment, we find that curiosity/input entropy and infogain correlate better with human similarity than task reward does. This provides an additional motivation to use input entropy for evaluation: it appears to be a better proxy for similarity to human behavior than task reward is.
>
> > If the idea is to convey the message that humans display curiosity as measured by your interpretation and way of measuring it, then there is a large body of text on human curiosity that already discusses these topics. Additionally, for this you don't need to train artificial agents.
>
> “Task reward with curiosity better explains human behavior than task reward alone” was meant to refer to correlations with human similarity; we did not intend to state a hypothesis about the motivation of humans. We have clarified the wording to resolve this confusion.
>
> > In addition, as discussed in the notes below, the empowerment and info gain the simplistic way that they are implemented are not actually good measures as a random agent is able to score strongly on those without having any intelligence.
>
> Regarding the high information gain obtained by the random agent, we have found that sharing discretization thresholds across agents within each environment improved results as it means that the Dirichlet distribution used for infogain is over the same support for each agent, resolving an inconsistency previously present. The random agent now achieves the highest infogain only in Minecraft, where it also achieves comparatively high reward.

---

> > ### Comment · AnonReviewer2 · 2020-11-24
> > **re:**
> >
> > Thanks for the notes and changes. However, still I'm not really sure what are the outcomes that you want the reader to take home.
> >
> > > comparing a set of agents based on curiosity measures an aspect of their behavior distinct from that measured by task reward.
> >
> > While I get this, the part that is not clear to me is that why one would want to do that? What is the utility that you want to get out of it? What are the scenarios that if one does this, there's some insight that you would get out of it that is useful? If there's a task reward, you are more or less are interested in optimising that. If the task reward is sparse, you are likely to use intrinsic objectives to be again better at the task. And if it's zero extrinsic reward, then intrinsic is all you are optimising.
> >
> > So, what is there to be gained from these measures that are not optimised but only measured? Can you give an actual example where a decision in terms of let's say which model to choose or which optimisation algorithm to use or anything that you suggest will be done based on these measures that are not captured by either the extrinsic or intrinsic rewards the agent is trained with?
> >
> > I hope I'm clear in conveying where I think the paper for me needs more work. As I mentioned earlier, it's an important area of research, so would like to see work like this to be more impactful.
> >
> > ---
> >
> > Some of the other comments have not been addressed.
> >
> > Additional comment:
> > - There are repetition in abstract that you can remove: "We find that all three objectives correlate more strongly with a human behavior similarity metric than with task reward. Moreover, input entropy and information gain both correlate more strongly with human similarity than task reward does."

---

> > > ### Author Response · Authors · 2020-11-25
> > > **Measurement of metrics; correlations in abstract**
> > >
> > > Thank you for your response.
> > >
> > > > While I get this, the part that is not clear to me is that why one would want to do that? What is the utility that you want to get out of it? What are the scenarios that if one does this, there's some insight that you would get out of it that is useful? If there's a task reward, you are more or less are interested in optimising that. If the task reward is sparse, you are likely to use intrinsic objectives to be again better at the task. And if it's zero extrinsic reward, then intrinsic is all you are optimising.
> > >
> > > A long-term objective of many researchers is building intelligent agents that behave similarly to humans, and our results hint at the possibility that the reward-based RL paradigm by itself may not be the most efficient in practice for working towards this goal. We suggest that measuring and optimizing for information theoretic properties such as our considered metrics, rather than extrinsic rewards, may be a promising addition to reward-based evaluation for this line of research.
> > >
> > > > There are repetition in abstract that you can remove: "We find that all three objectives correlate more strongly with a human behavior similarity metric than with task reward. Moreover, input entropy and information gain both correlate more strongly with human similarity than task reward does."
> > >
> > > These are two distinct statements. In the first sentence we are saying that the correlation of, e.g., input entropy with human similarity is greater than input entropy with task reward. In the second sentence we are saying that the correlation of task reward with input entropy is stronger than task reward with human similarity.

---

### Official Review · AnonReviewer3 · 2020-10-29
**Initial review for Submission 1867**

**Rating:** 3
**Confidence:** 4

**Review:**

**Summary**

The goal of this paper is to improve our understanding of reward-agnostic metrics drawn from the literature through comparison with human behaviour and task reward. This paper compares two intrinsic reward methods against three baselines on three Atari environments on five metrics, including task reward, a simple metric for human similarity, and three information-theoretic assessments of aggregated observation counts drawn from the literature, which they call task-agnostic metrics. The authors report the correlation between the different metrics.

**Strengths and Weaknesses**

Constructing a comparative understanding of the many methods for exploration, intrinsic motivation, and curiosity is a vastly underdeveloped area. I think that this paper's goal is to do some of that work, which I see as a strength. However, the experiments are not appropriately designed to provide reliable results and the paper includes substantial errors in understanding the existing literature, and as the paper is essentially an empirical survey, appropriately representing the other literature is critical.

Visually inspecting Figure 4, it appears that the results would be completely different if the no-op agent was excluded (and to a lesser extent, the random agent). My concern is that these baselines are categorically different from the agents we are actually interested in and appear to strongly affect the results. For example, without the no-op agent, it appears that the correlation between Human Similarity and Empowerment would be much weaker, and might actually be negative.

The Human Similarity metric does not seem to be a meaningful metric for what it is designed to measure. This is of particular concern to me because much of the interpretation of the data relies on comparison with the Human Similarity metric, so using such a simplified metric doesn't seem sufficient. The Human similarity data only considers which observations an agent shares with the human data, without regard for how many times each one visits a particular state. A human might make exactly one observation in a given bucket, and an agent making only one observation in that bucket would receive the same score for it as an agent that returns to that state millions of times. The generalization between state observations created by the preprocessing seems like it can only exacerbate the issue.

A similar concern arises when looking at the curiosity metric. Using entropy of sensory input visitation as a metric measures uniformity of visits to states, rather than measuring the ability of the agent to visit as many states as possible. In particular, you can construct examples in which visiting a small subset of states with uniform frequencies results in higher performance on this metric than covering more states, but with less uniform distributions. In principle, most researchers designing algorithms to improve exploration algorithms would care about this distinction. Intuitively, actually visiting a state and ensuring that the agent has observed what is there is important for ensuring the agent can find the optimal parts of the world.

The use of the word curiosity in this paper is problematic overall. Using the word curiosity to refer to both a metric and a set of methods leaves quite a bit of room for confusion for the reader. In particular, the methods and their metric are not as closely related as the authors suggest in the paper. While the authors appear to have the misconception that methods like ICM and RND are designed to increase the entropy over observations (stated on page 6), this is not the case. Importantly, these rewards are designed to be consumable, so they eventually no longer shape the behaviour of the agent and the agent is left to pursue (typically external) goals. That could result in visit frequencies being highly non-uniform.

The word curiosity has been used in the realm of reinforcement learning to refer to many very different methods, not necessarily methods that measure probability under a trained density model, and it isn't appropriate to provide this blanket definition of the word curiosity without some language to tell the reader that the word curiosity is simply a shorthand in this paper, in particular, to refer to methods that fall under the given definition.

While this paper makes clear calls to the foundation of ideas from the literature that are employed in this paper (e.g., work on curiosity, information gain, empowerment, human performance on Atari, etc.), there is no discussion of related kinds of comparative work that already exists in the literature. Neither the literature comparing multiple intrinsic reward agents nor the literature comparing the exploratory behaviour of RL agents with that of humans is discussed.

**Recommendation**

I am recommending that this paper be rejected on the basis of lack of appropriate evidence for their claims and inappropriate use of language to describe curiosity, a word with a diverse history in the literature.

**Specific Examples of Issues**

The characterization "Curiosity encourages encountering rare sensory inputs, measured by a learned density model" (p. 1) does not capture the definition of curiosity used as a metric: "the cross entropy of future inputs under a density model trained alongside the agent" (p. 4)
The characterization is inherently contradictory, as if curiosity is "successful" what does it mean for a sensory input to be rare? The characterization might be better captured by a definition that requires visiting many states.

The Go-Explore algorithm by Ecoffet et al. (2019) is explicitly not an intrinsic motivation algorithm (for example, see the paragraphs devoted to contrasting Go-Explore with IM methods on page 2 of Ecoffet et al., 2019) and the paper provides little evidence of the empirical success of IM methods, so citing the paper for such evidence does not appear appropriate. "Despite the empirical success of intrinsic motivation for facilitating exploration ..." (p. 1)

**Additional Feedback (Here to help, not necessarily part of decision assessment)**

I found myself trying to come up with a more appropriate name for the metric you call curiosity, and I think that "Observation entropy" might capture the mathematical definition appropriately.

More data might improve the quality of the results of your experiments; if you are interested in including other intrinsic-reward methods into future experiments, a list of fifteen different intrinsic rewards is included in https://arxiv.org/abs/1906.07865

Can you clarify what preprocessing is done for the images fed to the agents? This information belongs somewhere prior to "We first convert the RGB images to grayscale as they were seen by the agents." (p. 3)

I can't find the definitions of A (likely the action set?) and X (likely the set of possible 8x8 discretized images?) (used on p. 3) and it would be helpful to have these notations defined explicitly.

"has enable agents" (p. 1) Typo.

"Atari Learning Environment" (p. 2) I this was meant to be "Arcade Learning Environment"

"task-agnostic metric" (p. 5) Typo.

"human similarity it correlates" (p 8) Typo.

"For this reason, intrinsic rewards (Burda et al., 2018b) or human demonstrations (Aytar et al., 2018) are important to succeed at the game." (p. 12) Rather than "are important" I would suggest "have been important" since there is no evidence that there doesn't exist some method of another category that succeeds in Montezuma's Revenge that hasn't been published yet.

"chooses one of a set" (p. 12) reads a little strangely, since the agent is choosing an action, not a set.

ICM is not designed to be a complete agent (as it "can potentially be used with a range of policy learning methods," Pathak et al., 2017, p. 16) and so the phrase "is an exploration agent" (p. 12) is not accurate. I understand that you are using a PPO agent augmented with ICM, following Burda et al. (2018a), but that would be helpful information to include in your description of the agents in the appendix (perhaps along with a reminder to the reader about where to find the OpenAI implementations that you are using).

In Appendix D, the explanation of ICM (p. 12) would benefit from explaining what learning algorithm/agent architecture is used to optimize the intrinsic (or intrinsic + extrinsic) reward, to parallel the description given for PPO.

---

> ### Author Response · Authors · 2020-11-18
> **Alternate human similarity implementation, input entropy rationale and naming**
>
> Thank you for your feedback! We have evaluated an alternate human similarity implementation using Jensen-Shannon divergence to address your concerns, and renamed curiosity to input entropy to improve clarity.
>
> > The goal of this paper is to improve our understanding of reward-agnostic metrics drawn from the literature through comparison with human behaviour and task reward. This paper compares two intrinsic reward methods against three baselines on three Atari environments on five metrics, including task reward, a simple metric for human similarity, and three information-theoretic assessments of aggregated observation counts drawn from the literature, which they call task-agnostic metrics. The authors report the correlation between the different metrics.
>
> Your summary touches on the main aspects of our paper, although we would like to clarify that our focus is on evaluating the three task-agnostic metrics with respect to task reward and human similarity, not on evaluating the seven agents we used to collect data.
>
> > The Human Similarity metric does not seem to be a meaningful metric for what it is designed to measure. This is of particular concern to me because much of the interpretation of the data relies on comparison with the Human Similarity metric, so using such a simplified metric doesn't seem sufficient.
>
> To address your question, we have computed the Jensen-Shannon divergence between the sets of states visited by the human player and RL agent, as an alternative human similarity implementation. We have included these results in Appendix C. We find that the human similarity metrics based on Jaccard and JSD have a correlation of 0.78 with each other. JSD correlates less strongly with task reward, but the two implementations share similar correlations with the three task-agnostic metrics. We conjecture that this is the case because in the high-dimensional environment we study, an agent rarely visits the exact same state twice even after the binning is applied. As a result, measuring overlap and measuring overlapping densities yields similar metrics.
>
> > For example, without the no-op agent, it appears that the correlation between Human Similarity and Empowerment would be much weaker, and might actually be negative.
>
> We agree that the no-op agent has a strong influence on the correlations between the metrics, but do not see this as a problem. Metrics that capture the level of intelligence of an agent should assign low values to a no-op agent. We deliberately included both random and no-op as naive agents to cover two extreme behaviors, that is, the minimum entropy and maximum entropy action distribution. Our experiments show that our metrics assign lower values to the no-op agent than to the random agent, and generally assign lower values to the random agent than to more sophisticated trained agents.
>
> > A similar concern arises when looking at the curiosity metric. Using entropy of sensory input visitation as a metric measures uniformity of visits to states, rather than measuring the ability of the agent to visit as many states as possible.
>
> An agent that explores a larger number of unique states spreads out its lifetime input distribution over a larger number of states, making it more uniform, and thus will tend to achieve a higher score according to our input entropy metric.
>
> To further investigate your question, we have computed our metrics while using the same binning across all agents in the same environment, rather than deciding the binning percentiles for each agent individually. This way, the number of unique states is the same for all agents and visiting two different states rather than visiting the same state twice must increase entropy. We find that the shared discretization slightly increases the correlations for curiosity and human similarity and moderately increases the correlations between infogain and human similarity.
>
> > The use of the word curiosity in this paper is problematic overall. Using the word curiosity to refer to both a metric and a set of methods leaves quite a bit of room for confusion for the reader.
>
> We agree that curiosity has also been used to describe a broader class of intrinsic objectives in the literature. To avoid confusion, we have renamed the metric to “input entropy” throughout the paper.

---

### Author Response · Authors · 2020-11-18
**Summary of Changes**

We thank all four reviewers for their constructive feedback! We have updated the paper to include an additional variant of our human similarity metric as suggested by reviewer 3, to include p-values as requested by reviewer 4, improved our information gain estimation as suggested by reviewer 2, and repeated our analysis on an additional environment as suggested by reviewer 4. We believe that this addresses all major issues raised by the reviewers and we would be happy to engage in further discussion if there are remaining points that we should address.

To study the relationship of intrinsic metrics in an open-ended environment, we have repeated our analysis on Minecraft, a complex 3D environment with procedurally generated terrain where the agent can collect and place objects. The correlations for Minecraft confirm our findings from the Atari games and demonstrate the scalability of our task-agnostic evaluation metrics to open-ended 3D environments. Interestingly, we find that in Minecraft, input entropy correlates more strongly with human similarity than task reward does. We have added this result to the paper.

To improve our estimator of the information gain, we have made two small changes to the implementation. Specifically, we share the binning parameters across all agents, and only count each unique transition once when computing the posterior Dirichlet entropy. As a result, information gain correlates more strongly with human similarity than task reward does, and almost as strongly as input entropy. This further suggests the usefulness of information gain as a metric to evaluate aspects of agent behavior that are not captured by typical task rewards.

---

### Decision · Program_Chairs · 2021-01-07
**Final Decision**

**Decision:**

Reject

**Comment:**

The reviewers agree that the paper, in its current form, is not strong enough to allow for publication.  There are specific weaknesses that need to be tackled: a better correlation study; a clearer relationship to existing literature (and improvement on the novelty); clearer, more precise use of descriptions.

The authors are encouraged to continue with their work and submit a more mature manuscript.